## [Transparent Peer Review file · Nature Communications]

Endothelial C-type Natriuretic Peptide / Guanylyl Cyclase-B Signaling Prevents Pulmonary Arterial Hypertension

Corresponding Author: Professor Koichiro Kuwahara

Version 0:

Reviewer comments:

Reviewer #1

(Remarks to the Author)

I have read with a lot of interest the paper entitled Endothelial C-type Natriuretic Peptide / Guanylyl Cyclase-B Signaling Prevents Pulmonary Arterial Hypertension by Pr Kuwahara and colleagues.

Overall the paper is well written easy to follow, I particularly really appreciated the summary at end of all results section which truly help the readers on the take home message.

The findings are almost exclusively mice based 2 models MCT pyrol and hypoxia in multiple K.O models. Findings are in support of the conclusion.

I have several comments and suggestions to the researchers.

- 1) authors should provide cardiac output measurements in all the mice.
- 2) The role on endothelin signaling and smad2/3 signaling should be confirmed in vivo by the usage of endothelin receptor antagonist and sotatercept. As if the mechanism is right the genetically modified mice should respond differently to these standard or care compare to wild type mice.
- 3) the vascular remodeling data set should complemented by quantification of cell proliferation and apoptosis. Indeed changes in vascular remodeling should be associated to greater proliferation of either endothelial and smooth muscle while reduction in vascular remodeling by decrease proliferation and increase apoptosis. This should be investigated.
- 4) the open chest quantification of RVSP is clear limitation compare to what is done clinically, authors should mention that.
- 5) co culture between endothelial cells and vascular smooth muscle cells should be performed in boyden chamber to clearly demonstrates the autocrine paracrine phenomenon described.
- 6) given that the paper is highly based on in vivo experimentation I strongly recommend to compare efficacy of the therapeutic intervention to current standard of care like sotatercept and macitentan. This will be a big plus. These approach have been proposed by the preclinical research guideline in PAH doi: 10.1161/CIRCRESAHA.117.312579 it will be important to cite them.

Reviewer #2

(Remarks to the Author)

This report utilizes experimental models of disease, genetically-modified mouse lines with cell-specific deletion of C-type natriuretic peptide (CNP) or guanylyl cyclase (GC)-B, and human pulmonary artery endothelial cells, to define a role for the CNP/GC-B pathway in pulmonary arterial hypertension (PAH). The principal finding is that release of CNP from the vascular endothelium acts in an autocrine manner to trigger GC-B activation and prevent the expression of pathogenic mediators know to underlie the pathogenesis of PAH.

Major points

1. Some of these data have been published previously by these authors, albeit with focus on the systemic circulation rather than the pulmonary vasculature; specifically, the result of endothelial CNP deletion and smooth muscle GC-B ablation on

systemic blood pressure and the expression/release of endothelin (ET)-1 from pulmonary vascular endothelial cells (DOI: 10.1161/HYPERTENSIONAHA.116.08219). Other groups have reported on the endothelial-specific deletion of CNP and GC-B on systemic hemodynamics (DOI: 10.1161/CIRCULATIONAHA.117.033383; DOI: 10.1172/JCI74281). Therefore, these aspects of the manuscript are not new and could be replaced by appropriate citations.

2. Only male animals were used in this study. This is a significant drawback for a number of reasons. First, there is an established sex difference in the incidence, severity and treatment response of PH in the human population (DOI: 10.1111/bph.12281). Second, there are clear sex differences in the biological activity of CNP in mice, as these authors and others have reported previously (DOI: 10.1161/HYPERTENSIONAHA.116.08219; DOI: 10.1172/JCI74281; DOI: 10.1172/jci.insight.160416). In fact, an adverse cardiovascular phenotype in endothelial GC-B knockout mice has been described and shown to be restricted to females (DOI: 10.3390/ijms25147800). Therefore, it is imperative to utilize female mice in this work. The sex of the individual(s) from whom the pulmonary endothelial cells originated should also be provided, if known.

3. It would be important to conduct further studies to evaluate a potential role for NPR-C based on published work in this area. For example, there are many reports detailing positive effects of NPR-C activation in the pulmonary circulation, and PH in particular (DOI: 10.1016/j.biopha.2017.07.027; DOI: 10.1152/ajplung.2000.279.3.L511; DOI: 10.1016/j.lfs.2020.118877; DOI: 10.1152/ajplung.00477.2021). CNP also been reported to inhibit pericyte proliferation and activation in lungs of PH patients via PI3K and FoxO3 (DOI 10.1038/s42003-024-06375-3). The only attempt to rule in or out a role for NPR-C in the present investigation was by using siRNA knockdown in human pulmonary artery endothelial cells and evaluating the mRNA abundance of ET-1, IL-6 and CCL2 (Fig. S5). However, these studies were only conducted under normoxic conditions in which knockdown of GC-B also had little or no effect (Fig. 5). It is also stated in the Discussion that NPR-C mRNA was undetectable, which is not aligned with the data presented nor published work. This lack of mechanistic insight into a possible role for NPR-C is particularly important because endothelial CNP null mice have raised blood pressure (Fig. S1) whereas deletion of GC-B in the endothelium or vascular smooth muscle does not result in hypertension (Fig.S4). This implies that CNP is regulating blood vessel tone and blood pressure through activation of NPR-C? Therefore, statements in the Introduction and Results sections surmising that NPR-C is predominantly a clearance receptor are at odds with data provided in the current study and previous work by these authors and others (DOI: 10.1161/HYPERTENSIONAHA.116.08219; DOI: 10.1172/JCI74281). In addition, endothelial deletion of CNP results in systemic hypertension but does not affect pulmonary artery pressure at baseline whereas in experimental PH endothelial GC-B deletion exacerbates the increased RVSP but not systemic pressure? This suggests that CNP/GC-B signaling is specific to the pulmonary circulation, despite the down regulated GC-B expression that is reported? How do the authors explain this dichotomous result, particularly since the experimental models are produced by pathogenic stimuli that are not pulmonary specific (that is, hypoxia and monocrotaline). What happens to the abundance of CNP, GC-B and NPR-C mRNA in the systemic circulation under similar conditions?

4. The work is strengthened by the use of two discrete experimental models of PH. However, the majority of the studies, and all murine work, have been conducted using hypoxia alone or monocrotaline. This is a drawback since neither results in the formation of plexiform lesions; obliterative clone(s) of apoptotic-resistant endothelial cells, that characterize the human disease (PMID: 21868504). Since the conclusions of the study emphasizes the actions of CNP/GC-B pathway in endothelial cells this may be an important mechanism that is being overlooked? The gold standard Sugen/hypoxia model has only been utilized in one set of experiments in rats, but ideally this should have been used throughout. The hypoxia model of disease is also more akin to Class III PH and therefore the focus should probably be expanded to PH, rather than restricted to the PAH subgroup.

5. The administration of CNP-53 was an odd choice, particularly when minipumps were used that would largely circumvent the issues with the short plasma half-life of CNP-22. The potency of CNP-53 is considerably reduced compared to CNP-22 (DOI: 10.1093/eurheartj/eh308.P2532). In addition, whilst the CNP-53 was effective in preventing the development of PH (administration was prior to the model induction) it is a prerequisite for such studies, which claim potential therapeutic benefit, for a reversal protocol to be undertaken (that is, administration following establishment of the disease). Previous work has also demonstrated a beneficial pharmacological action of CNP in experimental monocrotaline-induced PH (DOI: 10.1164/rccm.200404-455OC), although further studies have disputed this finding (DOI: 10.1016/j.lfs.2011.07.009). At very least these published data should be acknowledged and discussed.

6. The impact of the work would be enhanced considerably if cells or tissues from patients with PH could be analyzed for expressional changes in CNP, GC-B and NPR-C, or the measurement of plasma CNP concentrations, to show that similar changes occur in human disease. Can the authors provide any information in this regard?

7. The reductions in GC-B mRNA abundance produced by cell-specific deletion and siRNA knockdown are approximately 50% (Fig. S2). A similar change is produced by hypoxia or monocrotaline exposure (Figs. 65% and 85%, respectively) so it is surprising that genetic ablation results in a phenotype since these proteins are reduced in disease? Is the down-regulation of expression of CNP/GC-B exaggerated in experimental PH in the transgenic lines? Also, it is surprising that genetic deletion of GC-B from endothelial cells or vascular smooth muscle cells each results in approximately 50% deletion of total mRNA abundance since GC-B expression on fibroblasts is significant, has clear functional effects in the heart and vasculature, and might be expected to contribute to disease progression (DOI: 10.1172/jci.insight.160416).

8. Published work by these authors (DOI: 10.1161/HYPERTENSIONAHA.116.08219) and one of the major findings of the present study is that endothelial deletion of CNP results in upregulated expression of endothelial ET-1. As ET-1 is established to play a pathogenic role in PH and the disease is treated with endothelin receptor antagonists (DOI: 10.1164/rccm.2312017), it would be helpful to show that, functionally, a similar pathway underlies the actions of CNP/GC-B signaling in experimental PH.

Minor points:

1. The dose of CNP-53 is stated as 1 ug/hr but a normalization to body weight should be given. In addition, it is stated that CNP-53 has limited interaction with NPR-C. What experimental evidence is this assertion based on?

2. The Discussion is rather verbose and contains considerable repetition of the results. This could be reduced in length and

focused on potential underlying mechanisms, context in terms of existing data on the regulation of vascular tone by CNP, and the receptors involved.

9. In experiments using Rp-8-Br-cGMP (Fig. 6) a PKGi alone control should be included.

10. There are a number of instances in which changes are described to have occurred 'irrespective of genotype'. This is not the case. For example, BP in endothelial CNP knockout mice is increased compared to wild type in normoxia, hypoxia and following monocrotaline exposure?

11. Antibodies specifically recognizing GC-A and GC-B have been notoriously difficult to generate. More clearcut evidence for protein specificity of the GC-B antibody should be provided. The immunohistochemical analyses in Fig. S3 are not convincing?

Version 1:

Reviewer comments:

Reviewer #1

(Remarks to the Author)

No further comments

Reviewer #2

(Remarks to the Author)

I would like to commend the authors regarding the extensive effort, both experimentally and textually, that they have made to address my initial criticisms (and those of the other referees) in their revised manuscript. The only outstanding issue in my mind is the relative roles of GC-B and NPR-C in the context of PH. A recently published manuscript (Pharmacol Res. 2025;219:107870. doi: 10.1016/j.phrs.2025.107870) also evaluated the role(s) of CNP in PH and concluded that both GC-B and NPR-C are important, depending on the index of disease severity being studied; other reports have also identified critical roles for NPR-C (Biomed Pharmacother. 2017 Sep;93:1144-1150. doi: 10.1016/j.biopha.2017.07.027 & Int J Cardiol. 2019 Apr 15;281:172-178. doi: 10.1016/j.ijcard.2018.06.001). Therefore, to offer a balanced perspective, it would be important to embellish the discussion by citing and acknowledging this previous work and a potential role for NPR-C in the pathogenesis and treatment of PH, without detracting from this study's principal finding that CNP/GC-B signaling plays a prominent role.

**Response to Reviewers:**

**Response to Reviewer 1**

We sincerely appreciate Reviewer 1's thoughtful and constructive feedback, which
greatly contributed to improving the clarity and overall quality of our manuscript. We are
also grateful for their recognition of the manuscript's readability and the clear
presentation of results. In response to their comments, we performed several additional
experiments, including cardiac output measurements, therapeutic comparisons with
standard-of-care agents, and co-culture assays, which have further strengthened our
conclusions. Our detailed, point-by-point responses are provided below.

***Major points***

***1. Authors should provide cardiac output measurements in all the mice.***

We thank the reviewer for this important suggestion. Cardiac output (CO) was not
included in the original submission. In response, we performed additional transthoracic
echocardiography in mice from the MCTp and Hx models. CO was calculated using the
left ventricular outflow tract (LVOT) method (Todd EA et al., 2022). No significant
differences in CO were observed across genotypes, indicating that the exacerbation of
pulmonary hypertension (PH) in knockout mice was not attributable to systemic cardiac

dysfunction. These data are now included as Supplementary Fig. 5 and described in the
Methods.

**Location in manuscript:**

- • Results, page 16-17, lines 220-222: “cardiac output (CO) assessed based on
echocardiography did not significantly differ among genotypes in the MCTp
model under either non-PH or PH conditions” Page 17, lines 230-232: “CO
assessed based on echocardiography remained unchanged across all genotypes
under both non-PH and PH conditions”
- • Methods, page 43, lines 608-612: “Cardiac output (CO) was measured in mice
from the hypoxia and SuHx models using transthoracic echocardiography and
calculated using the left ventricular outflow tract (LVOT) method as previously
described (Todd EA et al., 2022).”
- • Supplementary Fig. 5: new figure showing CO data.

*2. The role on endothelin signaling and smad2/3 signaling should be*
*confirmed in vivo by the usage of endothelin receptor antagonist and*
*sotatercept. As if the mechanism is right the genetically modified mice*
*should respond differently to these standard or care compare to wild*
*type mice.*

We thank the reviewer for this insightful comment. In wild-type mice, pulmonary
expression of CNP and GC-B receptor are down-regulated in both Hx-induced and
MCTp-induced PH, which leads to the activation of endothelin-related signaling and
SMAD2/3 signaling. In CNP-ecKO and GC-B ecKO mice, endothelial expression is
lower, even before the induction of PH, which results in more severe PH. Thus, in CNP
ecKO and GC-B ecKO mice, levels of endothelin-related signaling and SMAD2/3
signaling are thought to be much higher. Therefore, in CNP ecKO and GC-B ecKO mice,
endothelin receptor antagonist (ERA) and sotatercept essentially work in the same way
as in wild-type mice. On the other hand, treatment of ERA or sotatercept in addition to
CNP reveals the role of endothelin signaling and SMAD2/3 signaling in the protective
effects against PH exerted by CNP. If the effect of CNP against PH solely relied on either
SMAD2/3 signaling or endothelin signaling, we should see no additive effect with either
one. To address this point, we conducted additional in vivo testing of combination
treatments using the SuHx model in wild-type mice. Pharmacological interventions
included sotatercept (a SMAD2/3 signaling modulator), macitentan (an endothelin
receptor antagonist) and CNP-53. Each agent was administered either as monotherapy or
in combination with CNP-53 via appropriate routes.

These experiments demonstrated that sotatercept and macitentan independently
ameliorated RVSP and right ventricular hypertrophy with therapeutic efficacy
comparable to that of CNP-53. Notably, combination therapy with either sotatercept or
macitentan plus CNP-53 resulted in a more pronounced therapeutic effect, though in
terms of % reduction of RVSP, the combined effects seem to be somewhat attenuated

compared to the effects of individual treatments. This suggests CNP/GC-B signaling that
protects against PH partially overlaps both sotatercept and ERA (revised Fig. 10d-g).
Collectively, these findings demonstrate that CNP-GC-B signaling targets both
SMAD2/3 signaling and endothelin signaling in protecting PH. On the other hand, the
combinatory effect of CNP with sotatercept or macitentan is thought to reflect the
multifaceted actions of CNP, which also demonstrates the clinical efficacy of CNP in
combination therapy with standard treatment. These results are shown in the revised Fig.
10d-g and described in the revised Results section on page 29-30 and the Discussion
section on page 32. Details in changed parts are shown below.

**Location in manuscript:**

- • Results, page 29-30, lines 413-427: new section describing sotatercept,
macitentan, and CNP-53 monotherapy/combo data (see Fig. 10).
- • Discussion, page 32, lines 450-453: expanded discussion of mechanistic
implications.
- • Methods, page 47, lines 681-686: *Reagents* subsection corrected and expanded to
list macitentan and sotatercept.
- • Fig. 10: new figure showing intervention results.

***3. the vascular remodeling data set should be complemented by***
***quantification of cell proliferation and apoptosis. Indeed changes in***

*vascular remodeling should be associated to greater proliferation of*
*either endothelial and smooth muscle while reduction in vascular*
*remodeling by decrease proliferation and increase apoptosis. This should*
*be investigated.*

We appreciate the reviewer's insightful suggestion regarding the mechanistic
underpinnings of vascular remodeling. To further investigate the contribution of
endothelial CNP/GC-B signaling, we performed additional in vitro and in vivo analyses
focusing on smooth muscle cell (SMC) proliferation and apoptosis.

In vitro, human pulmonary arterial endothelial cells (HPAECs) were transfected with
siRNA targeting GC-B (siGC-B) and exposed to either normoxic or hypoxic conditions.
Conditioned medium from hypoxia-exposed siGC-B-transfected HPAECs significantly
increased human pulmonary arterial smooth muscle cell (HPASMC) proliferation, as
assessed using CCK-8 assays, compared with control conditioned medium from hypoxia-
exposed HPAECs. Furthermore, Boyden chamber assays demonstrated that PASMC
migration is markedly enhanced when co-cultured with siGC-B-deficient HPAECs under
hypoxia compared to that with control HPAECs under hypoxia (revised Fig. 7a, c).

Consistent with these in vitro findings, in vivo histological analyses revealed enhanced
smooth muscle proliferation (Ki67 immunostaining) and decreased apoptosis (tunnel
staining) in pulmonary arteries from GC-B eCKO mice with PAH, compared to genetic
control mice with PAH. Importantly, CNP treatment reversed those pathological

alterations by reducing proliferation and partially restoring apoptosis among SMCs
(revised Fig. 7d-g).

Together, these results indicate that endothelial GC-B deficiency promotes PASMC
proliferation and survival both in vitro and in vivo, thereby contributing to pathological
vascular remodeling. Conversely, endothelial CNP/GC-B signaling exerts anti-
remodeling effects by restraining SMC proliferation and resistance to apoptosis. These
results are described in the revised Results section on page 25-26 and shown in revised
Fig. 7. Details in changed parts are shown below.

**Location in manuscript:**

- • Results, page 25-26, lines 361-365: description of new proliferation and
cytotoxicity experiments.
- • Methods, page 46, lines 658-668: details of Immunohistochemistry of Ki67 and
TUNEL staining, page 51-52, lines 743-753: details of siRNA transfection,
conditioned media experiments, CCK-8 assays.
- • Fig. 7: new figure illustrating Ki67, TUNEL and CCK-8 assay results.

***4. the open chest quantification of RVSP is clear limitation compare to***
***what is done clinically, authors should mention that.***

We appreciate the reviewer's important point. We acknowledge that open-chest RVSP
measurements differ from the closed-chest approaches more commonly employed in
clinical settings. Nevertheless, open-chest measurements remain widely used in
preclinical models of pulmonary hypertension and have been validated in numerous
studies. In our study, this approach reliably detected significant differences in RVSP
between groups, and we have now clearly acknowledged this methodological limitation
in the revised Methods section (lines 625-627), with appropriate references to prior work
(Provencher et al., *Circ Res* 2018; DOI: 10.1161/CIRCRESAHA.117.312579).

**Location in manuscript:**

- • Methods: page 44, lines 625-627: statement added to acknowledge the limitation
of open-chest RVSP measurements, with citation to relevant literature [64].

***5. co culture between endothelial cells and vascular smooth muscle cells***
***should be performed in boyden chamber to clearly demonstrates the***
***autocrine paracrine phenomenon described.***

We thank the reviewer for this important suggestion. To examine whether endothelium-
derived CNP acts in a paracrine or autocrine manner, we conducted transwell-based
migration assays using Boyden chambers. In this setup, human HPAECs with or without
GC-B knockdown (siGCB) were seeded into the lower chamber (bottom well), while
HPASMCs, also treated \pm siGCB, were seeded into the inserts (revised Fig. 7b). This
configuration enabled us to assess directional migration of SMCs toward ECs under
hypoxic conditions.
Our results demonstrated that ECs with GC-B knockdown significantly promoted
HPASMC migration compared to control ECs. In contrast, GC-B knockdown in SMCs
alone did not affect migratory activity, indicating that the observed effect was primarily
driven by endothelial CNP-GC-B signaling (revised Fig. 7c). These findings support our
hypothesis that endothelial CNP-GC-B signaling plays a critical role in maintaining
endothelial integrity and suppressing the release of pro-migratory factors acting on SMCs
in a paracrine manner. These results are described in the revised Results section on page
26 and shown in revised Fig. 7. Details in changed parts are shown below.

**Location in manuscript:**

- • Results: page 26, lines 366-369: new subsection describing Boyden chamber
assay and its outcomes.
- • Methods: page 52, lines 756-763: description of Boyden chamber assay protocol.
- • Fig. 7: new figure illustrating Boyden chamber migration results.

***6. given that the paper is highly based on in vivo experimentation I***
***strongly recommend to compare efficacy of the therapeutic intervention***
***to current standard of care like sotatercept and macitentan. This will be a***
***big plus. These approach have been proposed by the preclinical research***
***guideline in PAH doi: 10.1161/CIRCRESAHA.117.312579 it will be***
***important to cite them.***

We appreciate the reviewer's insightful suggestion to benchmark our therapeutic strategy
against current standard-of-care drugs for pulmonary arterial hypertension (PAH). In line
with that recommendation, we conducted additional experiments using the
Sugen5416/hypoxia (SuHx) model in control mice to evaluate the therapeutic efficacy of
CNP-53 compared to that of sotatercept and macitentan.

Our new data demonstrate that macitentan and sotatercept, when administered
individually, each significantly improved RVSP and right ventricular hypertrophy to a
degree comparable to CNP-53. Notably, combining CNP-53 with either sotatercept or
macitentan led to further additive improvements in these parameters, suggesting
that CNP/GC-B signaling exerts additive effects to SMAD2/3 or endothelin pathway
inhibition (revised Fig. 10). These findings demonstrate the multifaceted actions of CNP
and support the therapeutic potential of CNP/GC-B signaling not only as a standalone
intervention but also in combination with established PAH therapies. We have also cited
the relevant preclinical research guideline recommending such comparative studies

(Provencher et al., *Circ Res* 2018; doi:10.1161/CIRCRESAHA.117.312579). These
results are described in the revised Results section on page 29-30, Discussion section on
page 32 and shown in revised Fig. 10. Details in changed parts are shown below.

**Location in manuscript:**

- • Results: page 29-30, lines 413-427; Fig. 10
- • Discussion: page 32, lines 450-453
- • Methods: page 42, lines 597-598; page 47, lines 681-686

**Response to Reviewer 2**

We sincerely appreciate Reviewer 2's constructive and thoughtful feedback, which
greatly contributed to improving the clarity and overall quality of our manuscript. In
response to their comments, we performed additional analyses and made corresponding
revisions to the text and figures where appropriate. Our detailed point-by-point responses
are provided below.

***Major points***

***1. Some of these data have been published previously by these authors,***
***albeit with focus on the systemic circulation rather than the pulmonary***
***vasculature; specifically, the result of endothelial CNP deletion and***
***smooth muscle GC-B ablation on systemic blood pressure and the***
***expression/release of endothelin (ET)-1 from pulmonary vascular***
***endothelial cells (DOI: 10.1161/HYPERTENSIONAHA.116.08219).***
***Other groups have reported on the endothelial-specific deletion of CNP***
***and GC-B on systemic hemodynamics (DOI:***
***10.1161/CIRCULATIONAHA.117.033383; DOI: 10.1172/JCI74281).***
***Therefore, these aspects of the manuscript are not new and could be***
***replaced by appropriate citations.***

We thank the reviewer for drawing attention to our prior work on the endothelium-derived
CNP. As noted, in our earlier study (Nakao et al., *Hypertension* 2017) we investigated
the physiological role of this pathway in regulating systemic blood pressure and
endothelin-1 (ET-1) expression/release, focusing primarily on systemic circulation and
blood pressure regulation. By contrast, the present study addresses the pathological
context of pulmonary arterial hypertension (PAH) and provides novel insights into the
detailed mechanisms by which endothelial CNP and its receptor GC-B-dependent
signaling modulates pulmonary vascular remodeling, with a particular focus on
endothelial-smooth muscle cell interaction. To avoid confusion regarding novelty or
overlap, we have revised the Discussion section to clearly distinguish the present study
from our earlier work and included appropriate citations to prior studies addressing
systemic hemodynamics (*Circulation* 2018; *JCI* 2014). Details of the changed parts are
shown below.

**Location in manuscript:**

Introduction, page 8-9, lines 100-106: we now cite our earlier study [9] and other relevant
studies [10, 11].

***2. Only male animals were used in this study. This is a significant***
***drawback for a number of reasons. First, there is an established sex***

*difference in the incidence, severity and treatment response of PH in the*
*human population (DOI: 10.1111/bph.12281). Second, there are clear*
*sex differences in the biological activity of CNP in mice, as these authors*
*and others have reported previously (DOI:*
*10.1161/HYPERTENSIONAHA.116.08219; DOI: 10.1172/JCI74281;*
*DOI: 10.1172/jci.insight.160416). In fact, an adverse cardiovascular*
*phenotype in endothelial GC-B knockout mice has been described and*
*shown to be restricted to females (DOI: 10.3390/ijms25147800).*
*Therefore, it is imperative to utilize female mice in this work. The sex of*
*the individual(s) from whom the pulmonary endothelial cells originated*
*should also be provided, if known.*

We thank the reviewer for raising this important point regarding sex differences in PAH
and CNP signaling. We fully agree that sex is a critical biological variable in PH
pathophysiology and therapeutic responses, and we appreciate the reviewer's detailed
references supporting this issue.

To address this concern, we conducted additional experiments using female CNP ecKO
and GC-B ecKO mice in both the monocrotaline pyrrole (MCTp) and hypoxia (Hx)
models and directly compared the results with those obtained in the corresponding male
mice. RVSP and right ventricular hypertrophy (RV/LV+IVS ratio) were assessed through
catheterization and tissue morphometry. These experiments revealed that, within each
genotype, no significant sex-specific differences in disease severity were detected in

terms of RVSP and RV/LV+IVS ratios. In all genotypes examined, endothelial GC-B or
CNP knockout mice exhibited similarly exacerbated PH phenotypes compared to controls,
irrespective of sex. Based on these findings, and to maintain consistency with our
previous studies and reduce experimental variability, we chose to use male mice in the
main study.

Regarding the HPAECs used in the in vitro experiments, we sourced them from Lonza;
however, the sex of the donor is not provided by the manufacturer and therefore remains
unknown. For the HPASMCs used in the Boyden chamber and proliferation assays, the
donor sex was female, and we have now included this information in the Methods section.
These results are described in the revised Results section on page 17-18 and shown in
revised Supplementary Fig. 6. Details of the changed parts are shown below.

**Location in manuscript:**

- • Results, page 17-18, lines 235-240: additional female mouse data added
(Supplementary Fig. 6).
- • Methods, page 39, lines 551-554: donor sex of HPAEC donor sex stated as not
provided (page 48, lines 698-700): HPASMC noted (page 51, lines 737-738).
- • Supplementary Fig. 6.

*3. It would be important to conduct further studies to evaluate a potential*
*role for NPR-C based on published work in this area. For example, there*
*are many reports detailing positive effects of NPR-C activation in the*
*pulmonary circulation, and PH in particular (DOI:*
*10.1016/j.biopha.2017.07.027; DOI: 10.1152/ajplung.2000.279.3.L511;*
*DOI: 10.1016/j.lfs.2020.118877; DOI: 10.1152/ajplung.00477.2021).*
*CNP also been reported to inhibit pericyte proliferation and activation in*
*lungs of PH patients via PI3K and FoxO3 (DOI 10.1038/s42003-024-*
*06375-3). The only attempt to rule in or out a role for NPR-C in the*
*present investigation was by using siRNA knockdown in human*
*pulmonary artery endothelial cells and evaluating the mRNA abundance*
*of ET-1, IL-6 and CCL2 (Fig. S5). However, these studies were only*
*conducted under normoxic conditions in which knockdown of GC-B also*
*had little or no effect (Fig. 5). It is also stated in the Discussion that*
*NPR-C mRNA was undetectable, which is not aligned with the data*
*presented nor published work. This lack of mechanistic insight into a*
*possible role for NPR-C is particularly important because endothelial*
*CNP null mice have raised blood pressure (Fig. S1) whereas deletion of*
*GC-B in the endothelium or vascular smooth muscle does not result in*
*hypertension (Fig.S4). This implies that CNP is regulating blood vessel*
*tone and blood pressure through activation of NPR-C? Therefore,*
*statements in the Introduction and Results sections surmising that NPR-*

*C is predominantly a clearance receptor are at odds with data provided in*
*the current study and previous work by these authors and others (DOI:*
*10.1161/HYPERTENSIONAHA.116.08219; DOI: 10.1172/JCI74281). In*
*addition, endothelial deletion of CNP results in systemic hypertension but*
*does not affect pulmonary artery pressure at baseline whereas in*
*experimental PH endothelial GC-B deletion exacerbates the increased*
*RVSP but not systemic pressure? This suggests that CNP/GC-B signaling*
*is specific to the pulmonary circulation, despite the down regulated GC-B*
*expression that is reported? How do the authors explain this dichotomous*
*result, particularly since the experimental models are produced by*
*pathogenic stimuli that are not pulmonary specific (that is, hypoxia and*
*monocrotaline). What happens to the abundance of CNP, GC-B and*
*NPR-C mRNA in the systemic circulation under similar conditions?*

We thank the reviewer for raising important questions regarding the potential role of
NPR-C in pulmonary hypertension (PH) and vascular homeostasis. We acknowledge that
previous studies have reported vasoprotective effects of NPR-C activation, particularly
in systemic vascular beds, and we agree that further investigation into this receptor's
function is warranted.

We would like to correct an imprecise statement in the original Discussion: NPR-C
mRNA was detected in human pulmonary artery endothelial cells (HPAECs), as
confirmed by RT-PCR analysis, and was sufficiently expressed to enable effective

knockdown by siRNA. In HPAECs knockdown of GC-B significantly increased EDN1
and IL6 expression, even under normoxic conditions (Fig. 5c and d), which is in contrast
to NPR-C knockdown under the same conditions (Supplementary Fig. 7). We have
revised the manuscript accordingly in the Results section on page 21 and the Discussion
section on page 36-37.

Our present study primarily focused on dissecting the endothelial CNP-GC-B signaling
involved in pathological pulmonary vascular remodeling and PH. Mice with endothelial-
specific deletion of either CNP or GC-B exhibited significantly and similarly exacerbated
PH phenotypes following hypoxia or monocrotaline pyrrole exposure, whereas smooth
muscle-specific GC-B deletion had minimal impact on PH progression (Figs 1, 2 and 3).
In addition, CNP administration prevented or improved PH exclusively through
endothelial GC-B (Fig. 9g-j). These in vivo findings strongly support the dominant role
of the endothelial CNP-GC-B axis in regulating pulmonary vascular remodeling during
the development of PH. Importantly, the enhanced expression of ET-1 and other
downstream inflammatory mediators and alterations in the balance of SMADs signaling
in GC-B-deficient endothelial cells further support a mechanistic link between GC-B loss
and disease exacerbation.

As the reviewer noted, NPR-C has been implicated in the regulation of systemic blood
pressure, and we observed elevated systemic pressure in CNP ecKO mice but not in GC-
B ecKO mice (Supplementary Figs. 1 and 2). This may suggest that physiological
systemic vascular tone is regulated through CNP-NPR-C signaling, whereas the
pathological pulmonary vascular remodeling in PH is primarily dependent on the

endothelial CNP-GC-B pathway. As the reviewer points out, endothelial CNP-GC-B
appears not to regulate pulmonary arterial pressure under physiological conditions, where
PAEC and PASMC are thought to maintain a physiological and sedated phenotype. Under
pathological conditions, endothelial CNP-GC-B signaling initially plays a protective role
in maintaining endothelial integrity and suppressing the activation of PASMC by working
against several pathological stress signaling pathways, but when the pathological stress
persists for a long time or is strong, endothelial CNP-GC-B signaling is diminished,
which leads to the progression of pathological pulmonary vascular remodeling and PH.
The entire loss of endothelial CNP-GC-B signaling further exacerbates the disease
progression. This context-specific engagement of CNP-dependent signaling may help
explain the observed dichotomy in systemic versus pulmonary responses.

While our current data do not exclude a role for NPR-C in PH and the systemic circulation,
our genetic models and functional analyses strongly support the notion that endothelial
CNP-GC-B signaling is the principal regulatory mechanism underlying the pulmonary
vascular remodeling seen under the pathological conditions leading to PH. At the same
time, our findings do not exclude the possible contribution of NPR-C and GC-B expressed
outside endothelial cells and SMCs, such as fibroblasts or pericytes, which also may
contribute to pulmonary vascular regulation. Given the prior evidence of vasoprotective
NPR-C signaling, further studies will be required to delineate whether NPR-C contributes
directly to pulmonary vascular remodeling under pathological conditions, in addition to
its established role in systemic vascular tone. We have revised the manuscript accordingly

to clarify our interpretation and to correct the earlier description of NPR-C expression.

Details in changed parts are shown below.

**Location in manuscript:**

• Results, page 21, lines 295-297

• Discussion, page 36-37, lines 517-532

***4. The work is strengthened by the use of two discrete experimental***
***models of PH. However, the majority of the studies, and all murine work,***
***have been conducted using hypoxia alone or monocrotaline. This is a***
***drawback since neither results in the formation of plexiform lesions;***
***obliterative clone(s) of apoptotic-resistant endothelial cells, that***
***characterize the human disease (PMID: 21868504). Since the***
***conclusions of the study emphasizes the actions of CNP/GC-B pathway in***
***endothelial cells this may be an important mechanism that is being***
***overlooked? The gold standard Sugen/hypoxia model has only been***
***utilized in one set of experiments in rats, but ideally this should have***
***been used throughout. The hypoxia model of disease is also more akin to***
***Class III PH and therefore the focus should probably be expanded to PH,***
***rather than restricted to the PAH subgroup.***

We thank the reviewer for highlighting important considerations regarding the choice of
pulmonary hypertension (PH) models. We agree that no single animal model fully
replicates the complex pathology of human PAH, especially in mice. The Sugden/hypoxia
(SuHx) rat model is widely accepted as the most representative for severe PAH with
obliterative vascular remodeling, including plexiform-like lesions (Abe et
al., *Circulation* 2010; Vitali et al., *Pulm Circ* 2014).

In our original submission, we utilized the MCTp and hypoxia-induced models of PH in
mice, which are well-established and widely used in the field, including in studies
addressing the molecular mechanisms underlying the development of PH. In response to
the reviewer's concern regarding the limitations of these models, especially from a
translational viewpoint, we have now incorporated the SuHx mouse model into our
revised study (revised Fig. 10). We tested the therapeutic effects of CNP53 administration
and its effects when combined with sotatercept or macitentan in the Su/Hx mouse model.
This enabled us to evaluate disease progression and therapeutic effects in a more
clinically relevant setting. By combining multiple models, we aimed to enhance the
translational relevance and mechanistic robustness of our findings. Details in changed
parts are shown below.

**Location in manuscript:**

- • Results, page 29-30, lines 413-427: newly incorporated data from the SuHx
mouse model, presented in Fig. 10.
- • Methods, page 42, lines 595-598: added description of the experimental design
for the SuHx mouse model.
- • References: included relevant citations (Vitali et al., Pulm Circ 2014; Abe et al.,
Circulation 2010) to support model selection rationale. [16], [17]

*5. The administration of CNP-53 was an odd choice, particularly when*
*minipumps were used that would largely circumvent the issues with the*
*short plasma half-life of CNP-22. The potency of CNP-53 is considerably*
*reduced compared to CNP-22 (DOI: 10.1093/eurheartj/eh308.P2532). In*
*addition, whilst the CNP-53 was effective in preventing the development*
*of PH (administration was prior to the model induction) it is a*
*prerequisite for such studies, which claim potential therapeutic benefit,*
*for a reversal protocol to be undertaken (that is, administration following*
*establishment of the disease). Previous work has also demonstrated a*
*beneficial pharmacological action of CNP in experimental*
*monocrotaline-induced PH (DOI: 10.1164/rccm.200404-455OC),*
*although further studies have disputed this finding (DOI:*
*10.1016/j.lfs.2011.07.009). At very least these published data should be*
*acknowledged and discussed.*

We appreciate the reviewer's critical evaluation of our pharmacological approach using
CNP-53. We chose CNP-53 based on both physiological relevance and pharmacokinetic
considerations. CNP-53 is recognized as the major circulating form of CNP in vivo, as
originally described by Minamino et al. (Biochem Biophys Res Commun. 1993; PMID:
8250942), and it has been widely used in prior in vivo studies due to its greater resistance
to enzymatic degradation by neutral endopeptidase (NEP) (Wendt et al., J Pharmacol Exp
Ther. 2015; PMID: 25650377).

To assess its pharmacodynamic profile, we conducted additional in vivo experiments
comparing the bioactivities of CNP-22 and CNP-53. Control mice were subcutaneously
infused with either peptide for 7 days, after which urinary cGMP levels were measured
using an ELISA. As shown in the revised Supplementary Fig. 8, CNP-53 administration
led to a statistically significant increase in urinary cGMP compared to the vehicle control,
whereas CNP-22 showed only a modest, non-significant effect. These results suggest that
CNP-53 more effectively activates the GC-B signaling pathway in vivo. Furthermore,
earlier studies have demonstrated that continuous subcutaneous infusion of CNP-22
failed to produce physiological effects such as bone growth, which supports our choice
of CNP-53 for in vivo application (Yasoda et al., Endocrinology 2009). Together, these
findings reinforce the suitability of CNP-53 as a therapeutic candidate in our study.

We acknowledge the reviewer's concern that our pharmacological design primarily
reflects a preventive approach. Indeed, most of our experiments were intended to dissect
the mechanistic contribution of CNP/GC-B signaling during the early phase of disease
development. Nevertheless, in the revised study using the SuHx mouse model, we

administered CNP-53, macitentan and sotatercept during the normoxic phase following
an initial 3-week hypoxic exposure. This timing enabled us to examine the effects of these
agents after pulmonary vascular remodeling had already been initiated. While this may
not represent a classical late-stage reversal model, it does address the therapeutic potential
of these interventions beyond prophylactic use.

Finally, we appreciate the reviewer pointing out previous conflicting reports on the effects
of CNP in experimental PH (DOIs: 10.1164/rccm.200404-455OC;
10.1016/j.lfs.2011.07.009). These studies were cited and discussed in our original
submission, and we have now revised the relevant section to more clearly acknowledge
the controversy and place our findings in the appropriate context.

**Location in manuscript:**

- • Results, page 21-22, lines 299-304: new data comparing urinary cGMP levels
after CNP-22 vs. CNP-53 infusion; Supplementary Fig. 8.
- • Methods, page 48, lines 691-695: urinary cGMP assay protocol added.
- • Supplementary Fig. 8

***6. The impact of the work would be enhanced considerably if cells or***
***tissues from patients with PH could be analyzed for expressional changes***
***in CNP, GC-B and NPR-C, or the measurement of plasma CNP***

*concentrations, to show that similar changes occur in human disease.*

*Can the authors provide any information in this regard?*

We appreciate the reviewer's suggestion regarding the relevance of our findings to human
disease. To address this issue, we performed an analysis of a publicly available single-
cell RNA sequencing (scRNA-seq) dataset (GSE169471) derived from lung tissue
samples from patients with idiopathic pulmonary arterial hypertension (IPAH) and
control donors. Our re-analysis focused on endothelial cell populations and revealed a
significant reduction in the expression of CNP (*NPPC*) and GC-B (*NPR2*) in endothelial
cells from IPAH patients compared to controls. These results are now included in the
revised manuscript (see Fig. 8 and corresponding text) and provide further support for the
pathological relevance of impaired endothelial CNP/GC-B signaling in human PAH.

**Location in manuscript:**

- • Results, page 26-27, lines 376-387: new scRNA-seq analysis of *NPPC* and *NPR2*
expression in endothelial cells from IPAH vs control lungs; Fig. 8.
- • Discussion, page 31, lines 430-433: reference to human data to emphasize
translational significance.
- • Methods, page 53, lines 766-776: description of dataset re-analysis and
endothelial subset selection.
- • Fig. 8.

*7. The reductions in GC-B mRNA abundance produced by cell-specific*
*deletion and siRNA knockdown are approximately 50% (Fig. S2). A*
*similar change is produced by hypoxia or monocrotaline exposure (Figs.*
*65% and 85%, respectively) so it is surprising that genetic ablation results*
*in a phenotype since these proteins are reduced in disease? Is the down-*
*regulation of expression of CNP/GC-B exaggerated in experimental PH*
*in the transgenic lines? Also, it is surprising that genetic deletion of GC-*
*B from endothelial cells or vascular smooth muscle cells each results in*
*approximately 50% deletion of total mRNA abundance since GC-B*
*expression on fibroblasts is significant, has clear functional effects in the*
*heart and vasculature, and might be expected to contribute to disease*
*progression (DOI: 10.1172/jci.insight.160416).*

We appreciate the reviewer's thoughtful and detailed assessment of GC-B expression
dynamics and their relationship to phenotype severity.

First, we would like to clarify that the ~50% reductions in GC-B mRNA shown in
Supplementary Figs. 2 and 3 reflect measurements from whole lung tissue made using
RT-PCR. These values are not cell-type-specific and instead represent the combined
expression across all GC-B-expressing compartments, including endothelial cells, SMCs,
and fibroblasts. Therefore, the ~50% reduction observed in endothelial-specific or smooth
muscle-specific GC-B knockout mice is consistent with the partial loss of GC-B from one

major cellular compartment, while other cell types retain expression. From this point of
view, it is noteworthy that the loss of GC-B from endothelial cells results in a 50% loss
of GC-B in the lung and that the loss of GC-B in SMCs results in 50% loss of GC-B in
the lung, suggesting that under physiological conditions, where fibroblasts may not be
activated and proliferating, GC-B in the lungs is supposed to be predominantly expressed
in the endothelial cells and vascular smooth muscle.

Likewise, the observed reductions in GC-B mRNA following hypoxia or MCTp exposure,
which were also measured at the whole-lung level, likely reflect disease-induced
downregulation across multiple cell types, including endothelial cells and SMCs.
Supporting this notion, our single-cell RNA-seq analysis of human IPAH lungs
(GSE169471) revealed a significant reduction in *NPR2* expression specifically within the
endothelial compartment (Fig. 8). These findings collectively suggest that under
pathological conditions, conditional knockout (cell type-specific deletion) and disease-
induced suppression, which may occur across multiple cell types, may act additively or
synergistically, leading to more pronounced functional loss of CNP-GC-B signaling. This
likely accounts for the more severe PH phenotype observed in endothelial-specific GC-B
knockout mice exposed to hypoxia or MCTp.

As the reviewer points out, fibroblast-derived GC-B may also contribute to
cardiovascular remodeling (DOI: 10.1172/jci.insight.160416), and its relative impact on
whole-lung reductions of GC-B mRNA cannot be excluded. However, in the context of
our current genetic models and phenotypic analyses, endothelial GC-B signaling

consistently emerged as the critical determinant restraining pulmonary vascular
remodeling and disease progression in PH.

These findings are described in the revised Results section on page 14 and Discussion
section on page 37-38. The new single cell RNA-seq data are shown in revised Fig. 8.

Details of the revised text are shown below.

**Location in manuscript:**

• Results, page 14, lines 185-188: clarified interpretation of GC-B mRNA
reductions at the whole-lung level.

• Discussion, page 37-38, lines 533-543: discussion of fibroblast GC-B with
citation.

• Supplementary Figs. 2, 3: unchanged; clarified interpretation in the legend.

• Fig. 8: new single-cell RNA-seq data showing *NPR2* downregulation in
endothelial cells.

***8. Published work by these authors (DOI:***

***10.1161/HYPERTENSIONAHA.116.08219) and one of the major***

***findings of the present study is that endothelial deletion of CNP results in***

***upregulated expression of endothelial ET-1. As ET-1 is established to***

*play a pathogenic role in PH and the disease is treated with endothelin*
*receptor antagonists (DOI: 10.1164/rccm.2312017), it would be helpful to*
*show that, functionally, a similar pathway underlies the actions of*
*CNP/GC-B signaling in experimental PH.*

We thank the reviewer for highlighting the importance of mechanistically linking
CNP/GC-B signaling to ET-1, a key pathogenic factor in PH. As noted, our earlier work
(Nakao et al., *Hypertension*2017) demonstrated that endothelial deletion of CNP
increases ET-1 expression and secretion under physiological conditions. To functionally
explore this relationship in the context of PH, we extended our current study by evaluating
the potential additive or synergistic effects of CNP-53 in combination with the endothelin
receptor antagonist macitentan. Specifically, in the SuHx mouse model, where ET-1
signaling plays a prominent role, we administered macitentan alone or together with
CNP-53 during the normoxic recovery phase following hypoxia exposure. Our findings
demonstrate that combination therapy with CNP-53 and macitentan resulted in
significantly greater improvements in pulmonary hemodynamics and right ventricular
hypertrophy than with either treatment alone. This suggests that CNP/GC-B signaling
likely exerts complementary or partially overlapping protective effects with endothelin
pathway inhibition, supporting the idea that modulation of ET-1 may represent one of the
functional mechanisms by which CNP/GC-B protects the pulmonary vasculature.

**Location in manuscript:**

• Results, page 29-30, lines 413-427: new data on combination therapy with CNP-53 and
macitentan in SuHx model.

• Discussion, page 35-36, lines 503-510: expanded interpretation on interplay between
CNP/GC-B signaling and ET-1.

• Figure: Added new data as Fig. 10.

*Minor points*

*1. The dose of CNP-53 is stated as 1 ug/hr but a normalization to body*
*weight should be given. In addition, it is stated that CNP-53 has limited*
*interaction with NPR-C. What experimental evidence is this assertion*
*based on?*

We thank the reviewer for this helpful comment. The CNP-53 dose has been corrected
and expressed as a normalized value of 0.2 mg/kg/day in the revised manuscript.

Regarding the statement that CNP-53 has limited interaction with NPR-C, we agree that

this was not appropriately substantiated and have revised the text accordingly to remove

this unsupported claim.

**Location in manuscript:**

- • Results, page 27, lines 390-392; page 28-29, lines 408-411; page 29, lines 415-
418.
- • Methods, page 47, lines 674-676: CNP-53 dose corrected to 0.2mg/kg/day.

***2. The Discussion is rather verbose and contains considerable repetition***
***of the results. This could be reduced in length and focused on potential***
***underlying mechanisms, context in terms of existing data on the***
***regulation of vascular tone by CNP, and the receptors involved.***

We thank the reviewer for this constructive feedback. We agree that portions of the
Discussion were redundant and have carefully revised this section to reduce repetition
and improve clarity. In the revised manuscript, we have focused the Discussion more
tightly on the underlying mechanisms of CNP/GC-B signaling in pulmonary vascular
remodeling and placed our findings in the context of existing literature on vascular tone
regulation and natriuretic peptide receptor function.

**Location in manuscript:**

- • Discussion, page 31-38: revised to reduce repetition and focus on mechanisms
and context.

***9. In experiments using Rp-8-Br-cGMP (Fig. 6) a PKGi alone control***
***should be included.***

We thank the reviewer for this helpful suggestion. We agree that inclusion of a PKG
inhibitor (PKGi)-alone control is necessary for appropriate interpretation. Accordingly,
we have repeated the experiment to include a PKGi-alone group (Rp-8-Br-cGMP without
CNP treatment), and the updated results have been incorporated into the revised Figure
6. These new data confirm that the observed effects of CNP-53 are dependent on PKG
signaling, as the PKGi alone did not reproduce the effects seen with CNP-53 treatment.

**Location in manuscript:**

- • Results, page 23, lines 320-329: updated with PKGi-alone data.
• Fig. 6: revised to include PKGi-alone group.

***10. There are a number of instances in which changes are described to***
***have occurred ‘irrespective of genotype’. This is not the case. For***
***example, BP in endothelial CNP knockout mice is increased compared to***
***wild type in normoxia, hypoxia and following monocrotaline exposure?***

We appreciate the reviewer’s attention to this issue. We agree that the phrase “irrespective
of genotype” was used imprecisely in several instances. In the revised manuscript, we
have carefully reviewed all such statements and corrected the language to more accurately
reflect genotype-dependent differences. For example, we clarified that systemic blood
pressure is elevated in endothelial CNP knockout mice compared to wild-type controls
under all tested conditions (normoxia, hypoxia, and monocrotaline), and that this reflects
a genotype-specific effect.

**Location in manuscript:**

- • Results and Discussion, pages11-38: revised wording to clarify genotype-
dependent effects.

***11. Antibodies specifically recognizing GC-A and GC-B have been***
***notoriously difficult to generate. More clearcut evidence for protein***
***specificity of the GC-B antibody should be provided. The***
***immunohistochemical analyses in Fig. S3 are not convincing?***

We appreciate the reviewer’s concerns regarding the specificity of the GC-B antibody.
We recognize the challenges associated with distinguishing between GC-A and GC-B
due to their sequence similarity (61.99% homology, as confirmed by the manufacturer).

To validate the specificity of the GC-B antibody, we performed siRNA-mediated
knockdown of GC-B (siGCB) in HPAECs. RT-PCR confirmed a marked reduction in
*NPR2* mRNA levels. Consistent with that finding, Western blot analysis showed a
corresponding decrease in GC-B protein levels in siGCB-treated cells. Importantly, the
signal detected by an NPR-A antibody remained unchanged under these conditions,
suggesting minimal cross-reactivity and supporting the specificity of the GC-B antibody.
These validation data reinforce the reliability of our immunodetection results.

**Location in manuscript:**

- • Results, page 15, lines 190-198: validation of GC-B antibody specificity.
- • Supplementary Fig. 4: updated with siRNA knockdown and Western blot
validation data.

Response to reviewers:

*Reviewer #2 (Remarks to the Author):*

*I would like to commend the authors regarding the extensive effort, both*
*experimentally and textually, that they have made to address my initial criticisms (and*
*those of the other referees) in their revised manuscript. The only outstanding issue in*
*my mind is the relative roles of GC-B and NPR-C in the context of PH. A recently*
*published manuscript (Pharmacol Res. 2025;219:107870. doi:*
*10.1016/j.phrs.2025.107870) also evaluated the role(s) of CNP in PH and concluded*
*that both GC-B and NPR-C are important, depending on the index of disease severity*
*being studied; other reports have also identified critical roles for NPR-C (Biomed*
*Pharmacother. 2017 Sep;93:1144-1150. doi: 10.1016/j.biopha.2017.07.027 & Int J*
*Cardiol. 2019 Apr 15;281:172-178. doi: 10.1016/j.ijcard.2018.06.001). Therefore, to*
*offer a balanced perspective, it would be important to embellish the discussion by citing*
*and acknowledging this previous work and a potential role for NPR-C in the*
*pathogenesis and treatment of PH, without detracting from this study's principal*
*finding that CNP/GC-B signaling plays a prominent role.*

We thank the reviewer for this thoughtful and constructive suggestion. In accordance with
the reviewer's recommendation, we have revised the Discussion to more fully
acknowledge the potential contribution of NPR-C to pulmonary hypertension. We now

cite the recently published study (Pharmacol Res 2025;219:107870) demonstrating that
both GC-B– and NPR-C–mediated actions of CNP may influence PH pathophysiology,
as well as earlier reports supporting a protective role of NPR-C signaling in experimental
PH and in PH secondary to left-heart disease (Biomed Pharmacother 2017;93:1144-1150;
Int J Cardiol 2019;281:172-178). These studies have been incorporated into the revised
manuscript, and the Discussion has been updated to provide a balanced perspective that
places our principal conclusion—namely, that endothelial CNP/GC-B signaling plays a
predominant role in the models tested—within the broader context of existing evidence
suggesting complementary roles for NPR-C.

**Location in manuscript:**

- • Discussion, pages 36–37 (lines 527–531)
- • References 59–61 added as requested.